# Systematic Modeling of Risk-Associated Copy Number Alterations in Cancer

**DOI:** 10.3390/ijms251910455

**Published:** 2024-09-27

**Authors:** Alejandra Guardado, Raúl Aguirre-Gamboa, Victor Treviño

**Affiliations:** 1Institute for Obesity Research, Tecnologico de Monterrey, Monterrey 64710, Nuevo León, Mexico; aleguamen_@hotmail.com; 2School of Medicine, Tecnologico de Monterrey, Monterrey 64710, Nuevo León, Mexico; 3Committee on Immunology, University of Chicago, Chicago, IL 60637, USA; raul.aguirre.gamboa@gmail.com; 4Section of Genetic Medicine, Department of Medicine, University of Chicago, Chicago, IL 60637, USA; 5oriGen Project, Tecnologico de Monterrey, Monterrey 64849, Nuevo León, Mexico

**Keywords:** survival models, TCGA, cancer prognosis, biomarkers

## Abstract

The determination of the cancer prognosis is paramount for patients and medical personnel so that they can devise treatment strategies. Transcriptional-based signatures and subtypes derived from cancer biopsy material have been used in clinical practice for several cancer types to aid in setting the patient prognosis and forming treatment strategies. Other genomic features in cancer biopsies, such as copy number alterations (CNAs), have been underused in clinical practice, and yet they represent a complementary source of molecular information that can add detail to the prognosis, which is supported by recent work in breast, ovarian, and lung cancers. Here, through a systematic strategy, we explored the prognostic power of CNAs in 37 cancer types. In this analysis, we defined two modes of informative features, deep and soft, depending on the number of alleles gained or lost. These informative modes were grouped by amplifications or deletions to form four single-data prognostic models. Finally, the single-data models were summed or combined to generate four additional multidata prognostic models. First, we show that the modes of features are cancer-type dependent, where deep alterations generate better models. Nevertheless, some cancers require soft alterations to generate a feasible model due to the lack of significant deep alterations. Then, we show that the models generated by summing coefficients from amplifications and deletions appear to be more practical for many but not all cancer types. We show that the CNA-derived risk group is independent of other clinical factors. Furthermore, overall, we show that CNA-derived models can define clinically relevant risk groups in 33 of the 37 (90%) cancer types analyzed. Our study highlights the use of CNAs as biomarkers that are potentially clinically relevant to survival in cancer patients.

## 1. Introduction

Cancer is the second-leading cause of death worldwide [1]. In clinical practice, prognostic risk assessment is important for optimizing treatment and informing patients and families regarding the severity of the disease [2]. The use of genomic-derived molecular biomarkers from cancer biopsies has been supported by several studies using The Cancer Genome Atlas (TCGA) [3] and the International Network of Cancer Genome Projects (ICGC) [4] cohorts. Moreover, transcriptional signatures, such as those of MammaPrint and Oncotype DX, are routinely used in clinical practice [5]. Meanwhile, other omics data types, such as copy number alterations (CNAs, also named somatic copy numbers), show stable signatures, functional effects, and a similar prognostic potential [6,7,8,9]. However, CNAs have not been adopted in clinical practice. Copy number data obtained from DNA arrays are attractive due to their affordability and genome-wide coverage, and their implementation and processing protocols are well-established and, in some instances, simpler than those of other types of omics.

Recent work on developing CNA prognostic models has been performed for some cancers, such as breast [10] and ovarian [11] cancer. However, the methods used to generate a prognostic signature are specific to each study. For example, one of them uses gene expression in addition to CNA [10], while the other uses bootstrapping to further select CNA regions and a representative gene from the regions [11]. These methodological differences complicate comparisons between signatures and across cancer types. Moreover, to our knowledge, there is no systematic analysis applying a homogenous pipeline to evaluate the prognostic value of copy number data in many cancer types.

Cancer heterogeneity is one of the greatest limitations of CNAs [12]. Due to chromosomal heterogeneity in cancer nuclei and among patients, CNA features tend to be very rare. This sparsity of CNAs complicates the appropriate modeling of clinically relevant time-to-event information. Nevertheless, we have recently shown that mutations, and presumably any other molecular alteration observed in a small number of individuals, can be properly associated with time-to-event data with methods such as VALORATE [13,14]. Another difficulty is the type of data used in copy number analyses, where the level of the alteration can be defined as −2 or −1 depending on the number of alleles lost, also called deletions, or +1 or +2 depending on the allele gain, also called amplifications. These discrete levels in a particular genomic region rarely fit a continuous model where increasing CNA levels generate an increasing risk (or decreasing risk) [15], limiting the modeling potential of CNA features.

Our underlying hypothesis was that CNA data carry prognostic information in all cancer types. Thus, to systematically evaluate the prognostic potential of CNAs for survival in cancer patients, we developed a uniform methodology that we applied to 37 cancer types from TCGA data. For this, we first performed survival analysis stratified by the level of alteration estimated per gene region, soft (−1, or +1) or deep (−2, +2), and the type of alteration, amplification, or deletion. Then, three methods were explored to generate risk groups based on the significantly associated gene regions. Therefore, our pipeline generates eight possible signatures per cancer type. We show that most cancer types can be divided into groups with significantly different risks. Moreover, the results of our detailed analysis of many cancer types show that CNAs add molecular information relevant to the prognosis that is independent of known clinical risk factors. Together, our results highlight the potential for use of CNAs as clinically relevant features associated with survival in cancer patients.

## 2. Results

### 2.1. Data and Model Building

We downloaded and curated a total of 11,158 unique samples across 33 cancer types and four additional sets from TCGA, which are summarized in Appendix A. As shown in Figure 1, for each cancer type, we obtained GISTIC files with a CNA estimation per gene of −1 and −2 for deletions, +1 and +2 for amplifications, and 0 for no alteration, as described in the Methods section. From these, four data modes were built: “Soft Deletions”, “Soft Amplification”, “Deep Deletions”, and “Deep Amplifications”. After removing CNAs present in fewer than four patients, each data mode was tested for its association with time to survival using the VALORATE test. The VALORATE test implements a more precise log-rank test that can identify associations with survival even in highly sparse features, a characteristic of CNAs in cancer.

We then used potential gene region associations to define low- and high-risk groups as described in the Methods section. In brief, we generated eight prognostic models using (1) soft deletions, (2) soft amplifications, (3) deep deletions, (4) deep amplifications, (5) MaxSums of soft deletions and amplifications, (6) MaxSums of deep deletions and amplifications, (7) Combinations of risk groups from soft deletions and soft amplifications, and (8) Combinations of risk groups from deep deletions and deep amplifications.

### 2.2. Survival Associations

For screening, we first associated each CNA univariately with survival using VALORATE. The general results for the associated regions are shown in Figure 2. All cancer types showed associations between CNAs and survival, but to different degrees. Overall, soft CNA deletions showed 25% greater associations (n = 119,286) than amplifications (n = 90,117). In contrast, for deep CNA, deletions showed 59% fewer associations (n = 7527) than amplifications (n = 18,210). Moreover, soft and deep associations did not seem to be correlated (Appendix A), suggesting that they carry different biological information; therefore, they could be independent and, potentially, they could be complementary. For example, amplifications in chromosome 1 for esophageal cancer (ESCA) showed 88 genes with deep CNA associations, while no gene associations were estimated from soft CNAs. This observation supports independent modeling at the soft and deep levels.

### 2.3. Signatures by Cancer Type

Eight survival models were built for each cancer type (Figure 3). Four of these models use a single data type (amplification or deletion) and mode (soft or deep). Two more models correspond to deep and soft models from the risk sum given by amplifications and deletions. The last two models also correspond to deep and soft models built by combining the amplification risk groups with those from deletions, as summarized in Figure 3, across cancer types. When compared with the corresponding soft models, deep deletions and amplifications were more significant in virtually all cancer types. Nevertheless, some cancer types, such as cholangiocarcinoma (CHOL), kidney renal clear cell carcinoma (KICH), thyroid carcinoma (THCA), and lymphoid neoplasm diffuse large B-cell lymphoma (DLBC), did not show enough deep amplifications or deletions. Thus, better models were defined from soft data for these cancer types. The two algorithms that used amplifications and deletions together showed greater significance than the corresponding models that used only amplifications or deletions. The MaxSum and Combinations models using deep data were usually more significant than those using soft data.

### 2.4. Clinical Relevance of CNA Signatures

To assess whether our approach could generate clinically relevant signatures, we ranked and evaluated all eight proposed models per cancer to define the most potentially useful models in a clinical setting (Figure 3B). We first ranked the models based on significance; those most significant tended to be those composed of the *MaxSum* and *Combinations* algorithms that use amplifications and deletions in the same model. Another factor we considered was whether the risk group could be distinguished. For this criterion, the *MaxSum* algorithm, which generates only three risk groups, was favored in contrast to the *Combinations* algorithm, which can generate up to nine risk groups that, in some cases, produce indistinguishable risk groups of a small number of samples (see Appendix A). The results are shown in Figure 4. The *Combinations* algorithm was selected for six cancer types (adrenocortical carcinoma, kidney chromophobe, kidney renal papillary cell carcinoma, pheochromocytoma and paraganglioma, thyroid carcinoma, and uveal melanoma, marked as ACC, KICH, KIRP, PCPG, THCA, and UVM, respectively). For these types of cancer, only four risk groups or fewer were needed instead of the nine possible combinations. On the other hand, the *MaxSum* algorithm was chosen for 26 cancer types, suggesting that it is the most efficient and simple algorithm. The use of a *single*-data model was selected for only five cases (cholangiocarcinoma, ovarian serous cystadenocarcinoma, rectum adenocarcinoma, testicular germ cell tumors, and uterine carcinosarcoma, marked as CHOL, OV, READ, TGCT, and UCS, respectively), mainly because those cancer types show a scarce number of associated alterations. However, the survival curves are different for most of these cancer types. In the case of ovarian cancer (OV), using only *deep deletions* was almost as significant as using *Combinations* but simplified by three risk groups only. In the case of de uveal melanoma (UVM), where the *Combinations* algorithm was chosen, only the high (amp)–high (del) combination was needed.

For 20 of the 26 cancer types for which the chosen model was *MaxSum*, the survival curves of the three risk groups (low, high, and no assignment) were significantly different (bladder urothelial carcinoma (BLCA), breast invasive carcinoma (BRCA), cervical squamous cell carcinoma and endocervical adenocarcinoma (CESC), colon and rectum adenocarcinoma (COREAD), colon adenocarcinoma (COAD), esophageal carcinoma (ESCA), glioblastoma multiforme and brain lower grade glioma (GBMLGG), head and neck squamous cell carcinoma (HNSC), kidney pan cancers (KIPAN), kidney renal clear cell carcinoma (KIRC), brain lower-grade glioma (LGG), liver hepatocellular carcinoma (LIHC), lung adenocarcinoma (LUAD), lung squamous cell carcinoma (LUSC), mesothelioma (MESO), sarcoma (SARC), skin cutaneous melanoma (SKCM), stomach adenocarcinoma (STAD), stomach adenocarcinoma and esophageal carcinoma (STES), and uterine corpus endometrial carcinoma (UCEC)). There were some cases where the *MaxSum* algorithm could generate only two risk groups (NA and low risk or NA and high risk). This may have occurred when there were no associations found in the screening for one risk group. Correspondingly, only two risk groups were generated for DLBC (low and NA), acute myeloid leukemia (LAML) (high and NA groups), and thymoma (THYM) (high and NA groups). For prostate adenocarcinoma (PRAD), the algorithm was able to assign both risk groups, but the low-risk and the NA groups had no death events and therefore showed equivalent survival curves. Similarly, for GBM, the survival curve of the high-risk group was comparable to that of the NA group.

All models were significant except for testicular germ cell tumors (TGCTs), in which the number of reported death events was low. For the majority of cancers, more than 20% of patients could benefit from an accurate CNA-based stratification of risk groups.

### 2.5. CNA Signatures Are Independent of Other Clinical Factors

The above analysis showed that the models generated by screening CNAs were capable of successfully identifying low- and high-risk groups. To explore whether the risk group assigned by the models could be useful in a clinical setting where common clinical cofactors were also available and used in risk assessment, we further analyzed whether the risk assessment signal from the CNAs was independent of the other available cofactors in the selected models shown in Figure 4.

As an example, Figure 5 shows the detailed analysis of the *MaxSum* model, which uses deep amplifications and deep deletions in lung adenocarcinoma (LUAD). The first analysis consisted of cofactor stratifications of the original model. The survival curves demonstrated that the low- and high-risk groups were clearly distinguished when stratified by nodule (N0, N1), tumor size (T1, T2, T3/T4), stage (i, ii, marginally iii/iv), age (<60, ≥60), or sex (Figure 5). These results suggested that the CNA-based low- and high-risk groups were independent of these clinical risk factors when analyzed individually. Next, we jointly modeled the CNAs with all available clinical risk factors using a multivariate Cox model. This multivariate model confirmed that the low- and high-risk groups were independent of the other clinical risk factors (Figure 5G). Moreover, the statistical significance of the low- and high-risk groups was greatest, which suggested that the estimated low- and high-risk groups for LUAD were more relevant than the other known clinical risk factors. Figure 6A shows a simplified representation of the CNA model.

Similarly, for breast cancer (BRCA), the model showed consistently significant risk groups in most stratifications (Figure 7). Figure 6B shows a simplified representation of the CNA model. We then performed cofactor stratification analysis for the other 12 cancer types from the selected models, as shown in Figure 4. A summary of this analysis is shown in Figure 8. Interestingly, the models stratified by clinical cofactors, in most cases, revealed at least one significant risk group for cancer of the bladder, breast, lung, colon, brain, head and neck, liver, skin, stomach, or uterine corpus.

Taken together, these results across cancer types suggest that the biological signal related to the survival of CNAs in cancer is independent of other known clinical risk factors for survival, and that the implementation of CNAs in the clinical setting could improve the risk assessment of patients with most cancer types.

## 3. Discussion

The search for cancer biomarkers has been an intense and long-lasting scientific labor in which most kinds of cellular molecules have been explored [16,17]. Nevertheless, the use of CNAs for prognostic models has not been fully exploited, though the corresponding methods are well-established and characterized. CNA stems from genomic instability, a hallmark of cancer, which can aid in the development of clinically relevant biomarkers for survival in cancer and potentially for other diseases.

We first discretized the data into *soft* and *deep* modes depending on the level of the CNA. Then, we *used amplification and deletion* modes independently or jointly. From these data, we propose three methods to generate prognostic models from CNA gene regions associated with survival. The three algorithms are (i) *single* data, which designate patients in risk groups showing the highest coefficient sum of risk-associated genes; (ii) *MaxSum,* an additive-like model assigning patients to the risk group whose sum of risk-associated genes is the largest; and (iii) *Combinations*, which combines risk designations from single-data risks of amplifications and deletions. We showed that MaxSum and Combinations worked well (for most cancer types). The *Combinations* method performed best for some cancer types. Nevertheless, the single-data method was useful in a few cancer types. Overall, the methods proposed in this study can stratify patient biopsies into high- and low-risk groups for most cancer types. Moreover, our study suggests that CNA data are suitable for generating prognostic models for most cancer types.

Most of our generated prognostic models define three or four risk groups, as we observed in 26 of the 33 cancer types analyzed; such types of models facilitate clinical interpretation. Nevertheless, some cases showed particularities. For lymphoid neoplasm diffuse large B-cell lymphoma (DLBC, n = 45) and uterine carcinosarcoma (UCS, n = 55), only two risk groups were generated, low and no risk, where the no-risk group became the high-risk group in practice. This presumably occurred due to the low number of samples, the low number of survival-associated markers, and the fact that all associations were linked to lower risk. A similar pattern was observed for pancreatic adenocarcinoma (PAAD, n = 180) and thymoma (THYM, n = 121), where only high- and no-risk factors were detected. Thus, no risk became lower risk in these patients. Among prostate adenocarcinomas (PRADs, n = 492), three risk groups were present, but the low-risk and no-risk groups seemed equivalent, presumably due to the low number of death events that occurred in the high-risk group. The three risk groups of glioblastomas (GBMs) were defined given the data and the model, but survival curves from high-risk and no-risk patients did not show clear differences. For paraganglioma and pheochromocytoma (PCPG), the combined model was used to generate four groups, but in practice, only two groups were distinguishable—one associated with a higher risk and the other with a lower risk. In contrast, for adrenocortical carcinoma (ACC), kidney chromophobe (KICH), kidney renal papillary cell carcinoma (KIRP), and thyroid carcinoma (THCA), the combined model generated three or four distinguishable risk groups. Overall, we were able to generate simple and clinically usable CNA signatures in most cancer types.

In this study, we aimed to provide a conceptual framework highlighting the potential of CNA-derived models as clinically relevant biomarkers for survival in cancer patients. Nevertheless, many approaches could be used to construct a prognostic signature from copy number data. In ovarian cancer, Graf et al. [11] followed a pipeline using a Cox model for screening in “*soft*” data and then chose the most significant gene as representative of the region to build a multivariate Cox model and split the linear prognostic score into terciles to generate risk groups. Although the pipeline is similar, our approach has many methodological differences. First, Graf et al. used only one modality of data, equivalent to the *soft* data we used here. Extending their work, we showed that *deep* data provide different information but are also more predictive. As a result, using *deep* data was best in 23 of the 37 cancer datasets. Thus, the implementation of *deep* types of CNAs is a substantial improvement. Second, we used four different models (amplifications, deletions, the MaxSum of amplifications and deletions, and Combinations of amplifications and deletions), providing another level of exploration. Overall, the best models used the *MaxSum* algorithm for 26 cancer types, but there were also 6 cancer types for which *Combinations* was used and 5 cancer types for which single data were used. Thus, the strategy of exploring different algorithms to combine information and provide better risk prediction is important. Third, the risk group in each model was generated by the total number of regions associated with each risk group, whereas Graf et al. used terciles of the linear prognostic score. One of the problems with Cox estimations in multivariate models is that colinear variables cannot be assessed and must be removed from the analysis. Indeed, Graf et al. chose a “reporter gene” showing the most significant univariate Cox value as representative to perform this operation and avoid the collinearity issue. Thus, their final model used the sum of the coefficients of 14 chosen “reporter genes”. In that model, it is not clear how the estimation is performed if a patient does not show an alteration in a reporter gene. In our implementation, the risk is assessed by the number of regions that are associated with risk groups, which avoids the collinearity problem and a possible dependency on the reporter gene. As we have shown here, our strategy works well in most cancers using copy number data.

We used VALORATE instead of the log-rank test or Cox model for gene region screening. Although in our internal records we noted differences with the log-rank test, we did not observe large differences in VALORATE or Cox estimations for these data types, suggesting that a Cox model could also be used instead of VALORATE, simplifying possible implementations.

One of the limitations of our study is that we used TCGA datasets, which are difficult to validate one-to-one because other, similar efforts, such as the International Cancer Genome Consortium (ICGC) data, represent different types of cancer and different survival times. Although the strategy worked in most cancer data used, it needs to be validated in specific cohorts.

We performed a stratification analysis of many cancer types (those showing many samples) by splitting the population based on the most reported cofactors. We showed that, in most cases, at least one CNA-derived risk group was significantly associated with risk, independent of other known clinical risk factors. Some risk groups for specific cofactors had a low number of samples, decreasing our statistical power. Nevertheless, in many of these cases, the tendencies were concordant with the original risk assigned by the model. Overall, the generated risk models are largely independent of common clinical risk factors, suggesting that CNA models provide novel biological signals implicated in cancer survival and can contribute essential risk information to clinical practice.

## 4. Methods

### 4.1. Copy Number Data

We downloaded TCGA data from Firebrowse (http://firebrowse.org, accessed on 1 December 2020), which is part of the Genomic Data Commons Data Portal (https://portal.gdc.cancer.gov, accessed on 1 December 2023). In particular, we used Clinical and Somatic Copy Number GISTIC 2 Level 4, from now on referred as CNA. Systematically, we used the data from http://gdac.broadinstitute.org/runs/analyses__2016_01_28/data/[CODE]-TP/20160128/gdac.broadinstitute.org_[CODE]-TP.CopyNumber_Gistic2.Level_4.2016012800.0.0.tar.gz, where [CODE] was replaced with the cancer type code. The CNA data were originally obtained by TCGA using microarray technology and preprocessed by GISTIC [18]. We used the data from “all_thresholded_by_genes.txt”. We used single and composited cancer types, bringing together 37 cancer datasets. These datasets are based on 33 cancer types. Nevertheless, some of them merge previously confusing types, such as GBMLGG composed of glioblastoma and low-grade glioma (GBM and LGG), COADREAD composed of colon and rectal adenocarcinomas (COAD and READ), STES composed of stomach and esophagus carcinomas (STAD and ESCA), and KIPAN composed of kidney cancers. Here, we also used these composited datasets for generality. Overall, we analyzed information from approximately 11,000 patients and more than 24,000 CNAs in gene regions.

### 4.2. Data Preprocessing

The data were already segmented by gene regions, containing the estimated somatic copy number per annotated gene, as determined by GISTIC [18], which corresponded to −2 for homozygous deletion, −1 for heterozygous loss, 0 for diploid, 1 for one copy gain, and 2 for higher-level amplification or multiple-copy gain. Here, the raw values were *C* = −2, −1, 0, +1, and +2, which corresponded to *C* = 0 as “normal”, *C* = −1 as “soft deletion”, *C* = −2 as “deep deletion”, *C* = 1 as “soft amplification”, and *C* = 2 as “deep amplification”. These values were preferred because they represent losses or gains more intuitively. Some studies have used a definition of CNA equivalent to soft [11]. Nevertheless, we also tested an extreme definition, *deep*, because it may represent genes that need to lose both alleles to become associated or genes that need more copies, impacting gene expression more drastically. We removed gene regions showing low recurrence, *C* <> 0, for fewer than four patients.

### 4.3. Alteration Modes for Association Screening

We used four modes of CNA alterations. We first distinguished amplifications from deletions. Then, we divide them by depth; we used “soft” when the test involved soft or deep alterations, and “deep” when only deep alterations were used. Thus, the mode “Soft Deletions” involved any deletion (*C* values of −1 or −2), and “Soft Amplifications” included any amplification (*C* values from +1 or +2), while “Deep Deletions” embraced only deep deletions (*C* values of −2) and “Deep Amplifications” comprised deep amplifications (*C* values of +2).

### 4.4. Screening for Associations with Survival

Only patients who showed valid survival information were considered. Only gene regions showing alterations in 4 or more patients were used. To test the association with survival, we generated two risk groups: patients showing a CNA alteration and those not showing a CNA alteration. The above risk groups were subjected to the classical log-rank test. However, we and others have demonstrated that using the log-rank test in heavily unbalanced groups is inappropriate because the assumptions are violated [13,19]. Therefore, we used VALORATE (v1.0.1), an R package published by our research group, to estimate the exact null distribution independent of the number of patients per risk group [14].

### 4.5. Building Prognostic Signatures

We used only the gene regions whose *p*-value from VALORATE was *p* ≤ 0.05. To generate prognostic signatures, we used three methods. The first method was performed for the four alteration modes individually (Soft Deletions, Soft Amplifications, Deep Deletions, and Deep Amplifications), separating those associated with higher risk and those associated with lower risk generating in at most three groups; the high-risk group, in which the patient shows a coefficient sum (∑β) of gene regions associated with greater risk than the ∑β of gene regions associated with lower risk; the low-risk group, in which the patient shows a ∑β of gene regions associated with lower risk than the ∑β associated with higher risk; and the no-risk group (NA), in which the patient does not show CNA associations or when there is a tie in which the patient does not show a preference for a low- or high-risk group. The second method used amplifications and deletions, summing the ∑β per risk group and designating the patient to the risk group showing a greater sum or otherwise designating the no-risk group as having ties or not having CNA associations. This method was run using *soft amplification* plus *soft deletions* or *deep amplification* plus *deep deletions*. The third method combined the risk group designations performed in the first method for amplifications with those designations for deletions, generating combinations from individual alteration modes. Depending on the combinations, this could generate up to 9 risk groups (H, L, and NA for amplifications combined with H, L, and NA for deletions). This method also involved the use of *soft* amplification combined with *soft* deletion or *deep* amplification combined with *deep* deletion. Thus, 8 models were built per cancer type.

## Figures and Tables

**Figure 1 ijms-25-10455-f001:**
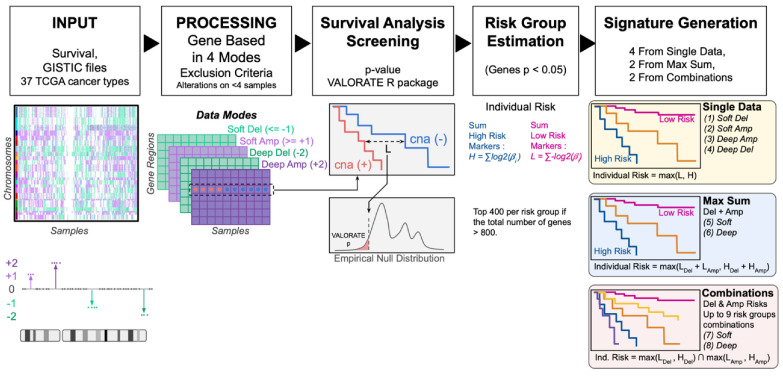
Systematic strategy. GISTIC files were processed and analyzed in 4 data modes. Survival screening was performed for each gene region. Significant high-risk gene regions were used to define a high-risk score per individual. The same approach was taken for low-risk gene regions. Individual risk was assigned to the higher score (H or L, if any), referred to as the single-data signature. Then, deletions and amplifications were used together, via MaxSum or Combinations. Ultimately, eight signatures were built for each cancer type.

**Figure 2 ijms-25-10455-f002:**
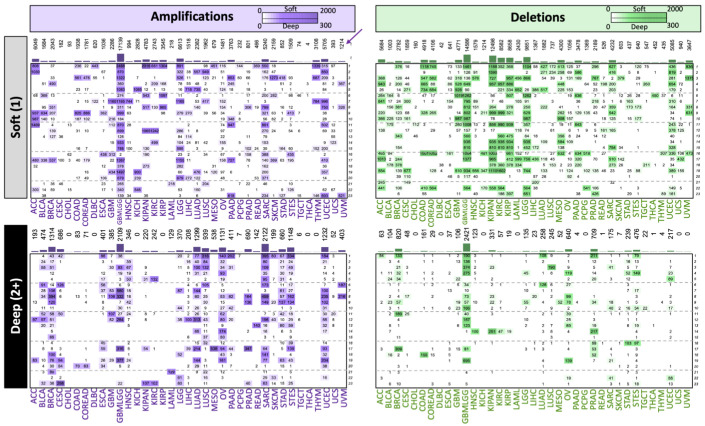
Gene region survival associations per data mode and cancer type. Each panel shows the number of gene regions associated with survival in the screening. Chromosomes are shown vertically, while cancer types are displayed horizontally. The bars and numbers at the top of each panel represent the sum of associations per cancer type. For instance, uveal melanoma (UVM, marked with an arrow) shows 1214 soft amplifications, which are distributed in chromosomes 4, 8, and 23.

**Figure 3 ijms-25-10455-f003:**
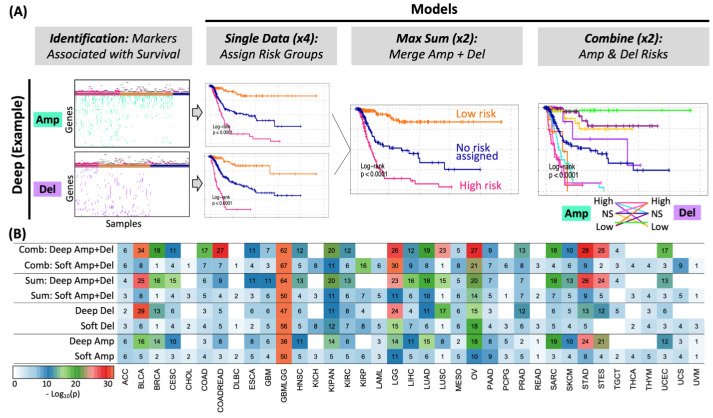
Overall model significance per cancer type. (**A**) Example of the generation of the four models from the deep data model. Briefly, gene regions associated with survival are used to assign risk groups in single-data models. Then, amplifications and deletions are mixed by MaxSum or Combinations. Four additional models are explored for the soft data mode. (**B**) The *p*-value of the log-rank test of each model (in rows) built for cancer types (in columns). *p*-values are shown in the logarithm base 10 scale. The number in each cell is rounded for simplicity. The colors are cut to a maximum of 30 for clarity.

**Figure 4 ijms-25-10455-f004:**
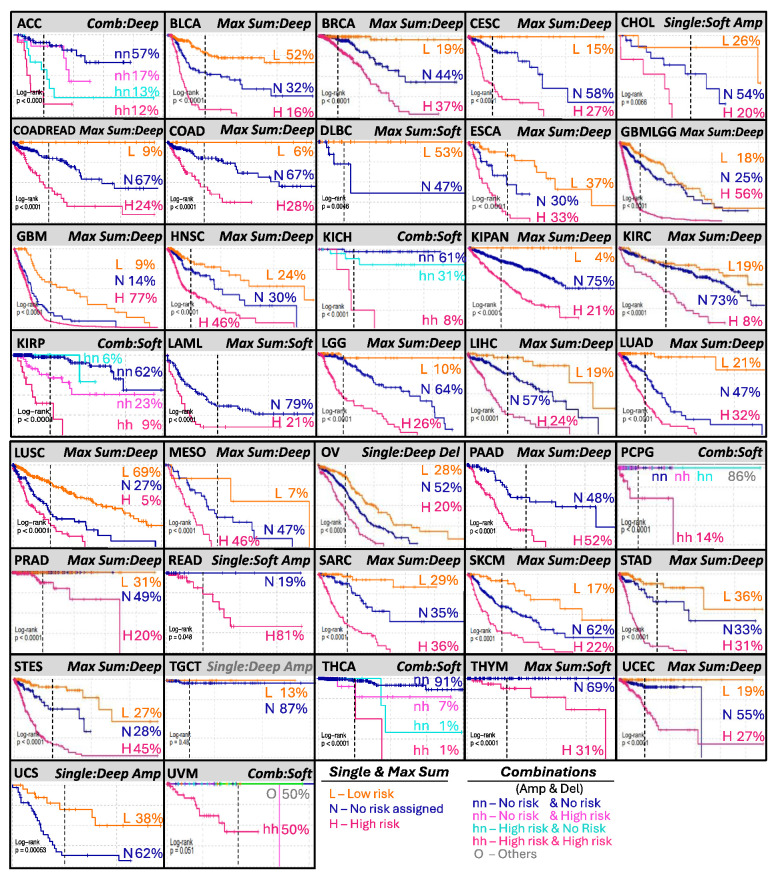
Selected signature per cancer type. Each panel shows the Kaplan–Meier curves of the risk groups generated by the selected algorithm and data used per cancer type. The percentages and colors represent the relative number of patients in each risk group. The vertical dotted line represents 1000 days (~3 years) for comparison.

**Figure 5 ijms-25-10455-f005:**
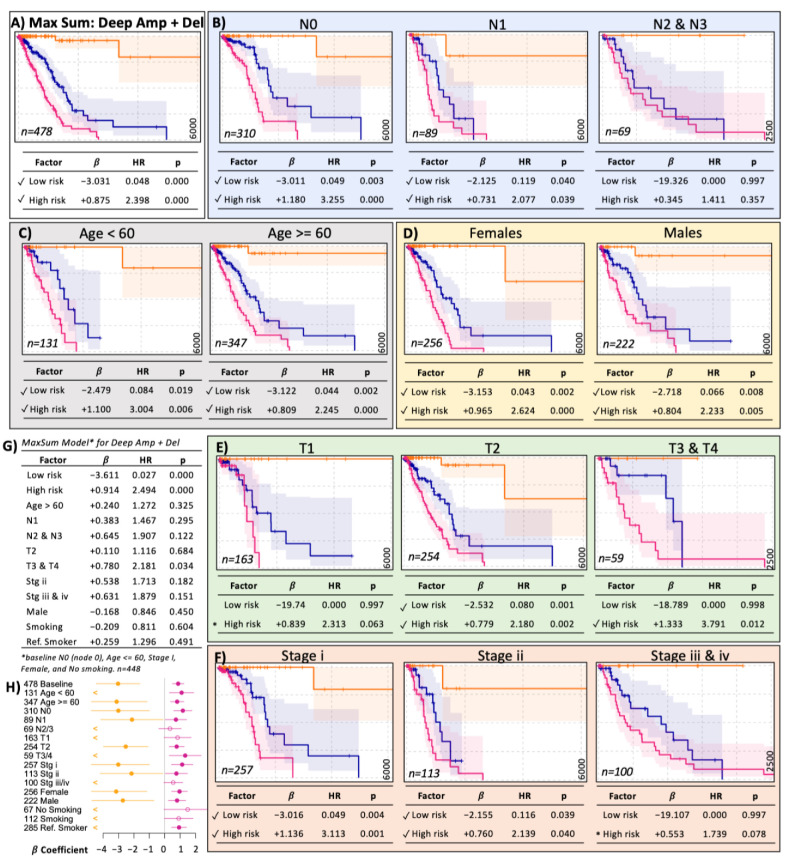
The MaxSum model for deep amplifications and deletions in lung adenocarcinoma (LUAD). (**A**) The LUAD model for deep CNA considers deep amplification and deletions for all patients (n = 478). Stratification of the model by nodules (**B**), age (**C**), sex (**D**), tumor size (**E**), and tumor stage (**F**). Most models show significant risk groups. (**G**) Multivariate statistical analysis of the model in (**A**) including common clinical cofactors. (**H**) Summary of the stratifications as a forest plot. The numbers on the left represent the sample size (n). Dots represent the *β*-coefficient estimation, closed for significant and open for not significant. Lines represent a 95% confidence interval. “<” refers to the lack of estimation in low-risk groups. A tick mark highlights significant risk estimation. “*” marks marginally significant risk groups. The frame colors group panels for clarity only. The vertical axis of survival curves ranges from 0 in bottom to 1 in top and horizontally from 0 in the left to 6000 days in the right except N2 & N3, T3 & T4, Stage iii & iv which are 2500 days.

**Figure 6 ijms-25-10455-f006:**
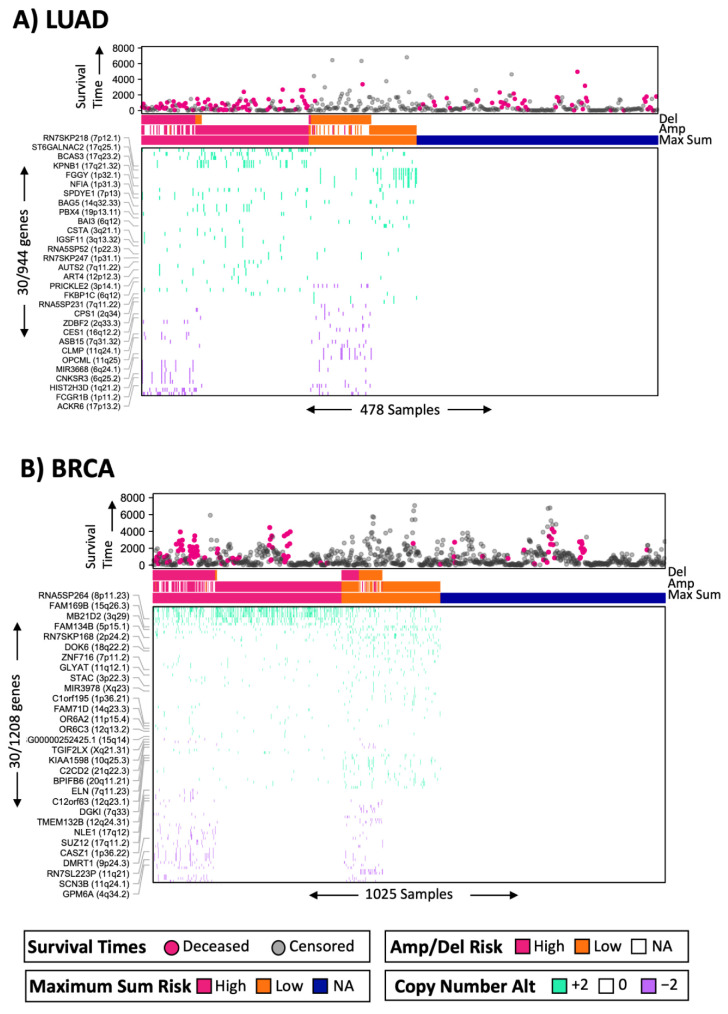
Simplified representation of the MaxSum models for deep CNA: (**A**) for lung adenocarcinoma (LUAD) and (**B**) for breast carcinoma (BRCA). Survival time is displayed in the top panel. Note that patients with shorter survival times and deceased tend to be shown at the left, in the high-risk groups. Del refers to the risk group assigned by the single-data model from deep deletions. Similarly, Amp refers to amplifications. MaxSum risk refers to the risk group assigned by the MaxSum model using data from both Del and Amp. The bottom panel show the CNA data, either amplification (+2 in green) or deletions (−2 in purple). Samples are shown on the horizontal axis. Genes are shown on the vertical axis. Only 30 representative genes were selected from different cytobands.

**Figure 7 ijms-25-10455-f007:**
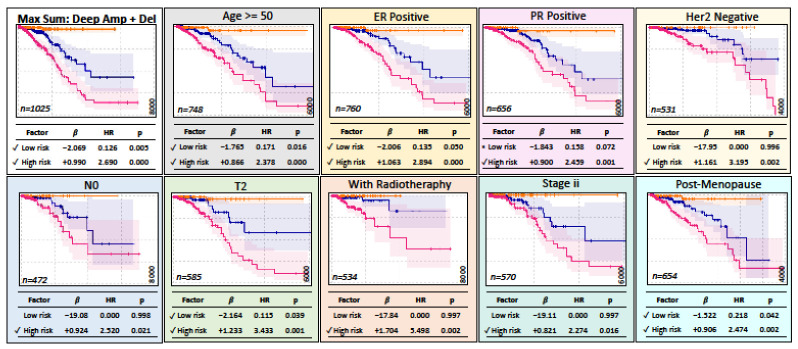
The MaxSum model for deep CNA with amplifications and deletions in breast carcinoma (BRCA). The top-left panel shows the model with all patients. The additional panels show the largest subpopulation within each clinical factor. A tick mark highlights a significant risk estimation. The frame colors are used for clarity only. The vertical axis of survival curves ranges from 0 in bottom to 1 in top and horizontally from 0 in the left to 4000, 6000, or 8000 days in the right as specified in right corner in each panel.

**Figure 8 ijms-25-10455-f008:**
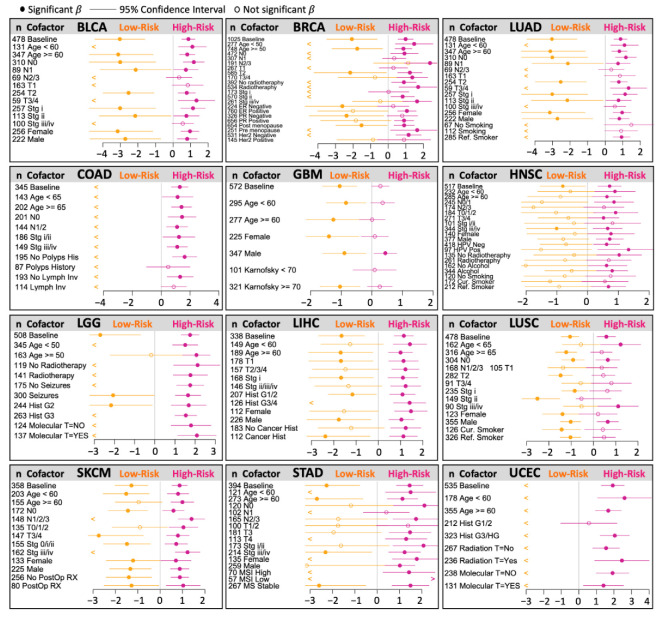
Cofactor stratifications of selected risk models. Each panel shows a “forest” plot summarizing the cofactor stratification of each model. The numbers on the left represent the sample sizes (n). Dots represent the *β*-coefficient estimation, closed for significant and open for not significant. Lines represent a 95% confidence interval. “<” refers to a lack of estimation in low-risk groups (commonly when there are no events). Most stratifications showed at least one significant risk group.

## Data Availability

The data used in this study were obtained from and are available at the TCGA data portal.

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
