# Peer review of "Systematic Modeling of Risk-Associated Copy Number Alterations in Cancer"

_ijms, 2024, doi:10.3390/ijms251910455_

Round 1

Reviewer 1 Report

Comments and Suggestions for Authors

In the study titled "Prognostic cancer signatures from copy number data," Guardado et al offers important insights into the prognostic potential of copy number alterations (CNAs) across various cancer types, but it requires substantial revisions to improve clarity and rigor, particularly in the methods and discussion section. The current description of the methodologies used for analyzing CNAs and constructing prognostic models lacks the necessary detail and transparency. Specifically, the procedures for data processing, classification of alterations into "deep" and "soft" categories, and the methods for combining single-data and multidata models need more thorough explanation. These aspects are crucial for understanding how the models were developed, validated, and how they can be replicated. Enhancing the detail and clarity in these areas will significantly strengthen the paper and its contributions to the field.

1.     The title "Prognostic Cancer Signatures from Copy Number Data" does not fully capture the scope and contributions of the study as described in the abstract. Given that your research involves a systematic analysis of copy number alterations (CNAs) across various cancer types, the title should reflect the comprehensive and multi-faceted nature of the study. Consider revising the title to better convey the depth of the analysis and the significance of the findings.

 2.    The introduction would benefit from a clearer explanation of the overarching hypothesis, as this is essential for understanding the context and significance of your study. While you mention recent work on CNA prognostic models in specific cancers like breast, ovarian, and lung, the introduction lacks a clear comparison of these studies. Furthermore, you refer to a "systematic analysis" of CNAs in multiple cancer types, but this term needs clarification. Many studies have investigated CNAs in real-world samples, so it is important to explicitly describe how your study differs from existing research. Please provide a clear explanation of what constitutes a systematic analysis in your context and how your approach offers novel insights or improvements over previous work.

 3.    How did you address potential biases in CNA data acquisition and processing across different cancer types and TCGA datasets? Detail any normalization techniques used to adjust for batch effects and technical variability across different datasets. For example, did you use any methods to standardize CNA calls from different platforms or preprocessing steps? Describe the quality control measures implemented to ensure that data from different sources or cancer types are comparable. Provide details on how you harmonized CNA data from different platforms or sequencing technologies. Were there any specific adjustments made to align data from diverse sources?

 4.    The classification scheme used in your study includes "Soft Deletions" (covering deletions with C values of -1 or -2) and "Deep Deletions" (covering only deep deletions with C values of -2). Since "Soft Deletions" already includes deep deletions (-2), could you clarify the rationale behind differentiating between "Soft" and "Deep" categories? Specifically, what is the purpose of having these two separate categories if there is an overlap? The same question applies to the classification of amplifications. Also, the exclusion criteria is not robustly explained. Also, what was the rationale of choosing those studies? This needs to be explained.

 5.    Many abbreviations have not been explained in the first instances.

 6.    Did you replicate your analyses in independent datasets or with different methodologies to confirm consistency? The figures are not self-explanatory,  legends and captions are not well defined. Rewrite the legends and captions for all the figures.

7.    The Discussion primarily revisits the results without sufficiently engaging with current literature to contextualize and strengthen the study's findings. It is important to compare your results with existing research to highlight how your study contributes to the field of CNA analysis. Additionally, this study should address how its findings could influence current clinical practices. Author need to cite this paper and elaborately discussed in discussion section related to NGS workflow and CNA detection. Gupta, V.; Vashisht, V.; Vashisht, A.; Mondal, A.K.; Alptekin, A.; Singh, H.; Kolhe, R. Comprehensive Analysis of Clinically Relevant Copy Number Alterations (CNAs) Using a 523-Gene Next-Generation Sequencing Panel and NxClinical Software in Solid Tumors. Genes 2024, 15, 396. https://doi.org/10.3390/genes15040396

  8.    Authors should include a section that outlines both the strengths and limitations of their study. Discuss how your model performs in the context of real-world samples and its potential applicability to clinical settings. This will provide a clearer understanding of the practical implications and utility of your research.

Reviewer 2 Report

Comments and Suggestions for Authors

This is a very interesting and innovative manuscript about the genomics of cancers and the role of CNVs. The authors call them CNAs, but they mean CNVs.

1. CNA is not a very good term. Sometimes CNAs are used to designate somatic CNVs, but in practice, such a distinction is unnecessary and even confusing. We do not have different names for SNV depending if it's somatic or germline. I recommend the author use CNV instead of CNA.

2. While the author analyses CNVs and discusses their impact on transcriptomic regualtion, they do not discuss transcriptomic profiles of cancer very much. This is a serious weakness, and the authors should really consider citing the papers that address the transcriptomic profiles. These articles give a very good overview of the transcriptomic changes (PMID: 25496518, 29050494, 29250102, 38966281).

3. Did the authors find any specific CNVs that are related to the survival of some specific cancers?

Reviewer 3 Report

Comments and Suggestions for Authors

The authors came back to an old finding that acquired copy number alterations (CNAs) show typical patterns in each specific tumor type. What is new is that they use a computer model for building prognostic signatures. The later is promissing and I have no concerns about this actual work authors present. Ths part seems well done and is convincingly presented.

My concerns only refer to previous spade work, which authors do not consider enough, and to some aspects, which should be also included which would strenghten their argumentation and also show some limitations.

For spade work: 

- already with the inclusion of metaphase-chromosome based CGH in 1992 in solid tumor research first tumor specific signatures and attempts for correlation of acquired CNAs with clinical outcome was done - look for works for Prof. E. Gebhart - e.g. https://pubmed.ncbi.nlm.nih.gov/?term=gebhart+cgh+tumor

For aspects, which should be also included which would strenghten their argumentation :

- besides there are several papers of Prof. T. Liehr's group, which recently showed repeatedly that murine solid tumor cell lines show the same CNA patterns (if they are translated into human genome) like the corresponding human tumors. - e.g. https://pubmed.ncbi.nlm.nih.gov/?term=liehr++tumor+cell+line+murine
This point is strenghtening the argument of the authors that CNAs are important for survival of solid tumors - and they are the same in mouse and human!

For studies showing limitations of the used approach:

- Finally, recently Ljubic et al. (PMID: 35205427) showed that satellite DNA-derived RNA is present in advanced solid tumors, a finding which was underscored by the finding (PMID: 38522124) that this (over-)expression is in parts due to satellite DNA amplification - it needs to be added as limitation of this present study that such alterations are technically not detectable by aCGH.

Round 2

Reviewer 1 Report

Comments and Suggestions for Authors

Authors responded all the queries.

Thanks

Reviewer 2 Report

Comments and Suggestions for Authors

All the concerns have been addressed.